# The Intestine of *Drosophila melanogaster*: An Emerging Versatile Model System to Study Intestinal Epithelial Homeostasis and Host-Microbial Interactions in Humans

**DOI:** 10.3390/microorganisms7090336

**Published:** 2019-09-09

**Authors:** Florence Capo, Alexa Wilson, Francesca Di Cara

**Affiliations:** Department of Microbiology and Immunology, IWK Research Centre, Dalhousie University, 5850/5980 University Avenue, Halifax, NS B3K 6R8, Canada

**Keywords:** *Drosophila melanogaster*, microbiota, intestinal epithelium, small intestine, inflammatory bowel disease, midgut, host-pathogen/commensal interactions, innate immunity, immunometabolism

## Abstract

In all metazoans, the intestinal tract is an essential organ to integrate nutritional signaling, hormonal cues and immunometabolic networks. The dysregulation of intestinal epithelium functions can impact organism physiology and, in humans, leads to devastating and complex diseases, such as inflammatory bowel diseases, intestinal cancers, and obesity. Two decades ago, the discovery of an immune response in the intestine of the genetic model system, *Drosophila melanogaster*, sparked interest in using this model organism to dissect the mechanisms that govern gut (patho) physiology in humans. In 2007, the finding of the intestinal stem cell lineage, followed by the development of tools available for its manipulation *in vivo*, helped to elucidate the structural organization and functions of the fly intestine and its similarity with mammalian gastrointestinal systems. To date, studies of the *Drosophila* gut have already helped to shed light on a broad range of biological questions regarding stem cells and their niches, interorgan communication, immunity and immunometabolism, making the *Drosophila* a promising model organism for human enteric studies. This review summarizes our current knowledge of the structure and functions of the *Drosophila melanogaster* intestine, asserting its validity as an emerging model system to study gut physiology, regeneration, immune defenses and host-microbiota interactions.

## 1. Introduction

The structure and function of the gastrointestinal (GI) tract are broadly conserved in metazoans [1]. The GI mediates the bulk of an organism’s nutrient absorption and has evolved as one of the largest and most dynamic organs of the body [2]. In this role, the GI tract is a central mediator of the metabolic status of an organism and also protects the organism against ingested biotic and abiotic hazards [3,4]. 

The GI absorbs and processes ingested food to fulfill the energy demands of an organism’s development, reproduction, and survival [5]. The GI tract is also a major source of neuronal and endocrine signals, which modulate food intake and nutrient storage and regulate the activity of other organs, such as the pancreas and the brain [6,7]. Additionally, the GI tract forms the largest and most important immune epithelial barrier that protects the organism against external dangers posed by ingested harmful pathogens. The GI tract maintains, under ideal circumstances, mutualistic and symbiotic relationships between a diverse and dynamic community of microorganisms and the host [8,9]. 

The intestinal epithelium is the single cell layer that form the lining of the intestine of the GI.

The intestinal epithelium has defense mechanisms to detect and eliminate invading pathogens. The intestinal epithelial cells (IECs) deploy several immune response strategies such as innate immune receptor families that recognize specific microbe-associated molecular patterns (MAMPs) and in turn activate cellular immune signaling strategies that include the production of microbicidal reactive anionic species and other antimicrobial molecules such as antimicrobial peptides [10,11,12]. These immune responses must be tightly regulated because a constant activation of the immune system (inflammation) may result in pathological conditions such as inflammatory bowel disease (IBD) and might lead to tumorigenesis [13,14]. Additionally, homeostasis between the microbiota and the immune system is critical for host physiology. For example, unbalanced changes in the diversity of the microbiota (dysbiosis) can impact growth and development during childhood and contribute to the pathogenesis of chronic metabolic diseases such as diabetes and obesity [15]. Moreover, the microbiota generates bioproducts, such as short-chain fatty acids (SCFAs), that may impact the physiology of the gut and distant organs such as the brain [16]. The impact of host–microbial interactions on the overall organism’s health is further highlighted by the correlation between dysbiosis and the prevalence of severe diseases, since disturbances in these interactions are associated with the insurgence of enteric diseases, such as IBD and neurodegenerative diseases, such as Alzheimer’s disease [17].

Overall, genetic and environmental factors (e.g., diet) can affect intestinal epithelial function, stability, and homeostasis and lead to severe pathologies. Dissecting the signaling pathways that modulate metabolic signaling, host–microbial interactions and stress responses in the gut epithelium is, therefore, pivotal for understanding how enteric health is achieved and maintained. Studies of model organisms have served as a great resource to untangle the complex mechanisms that determine the health of the intestinal epithelium in humans. 

The *Drosophila* genome has been fully sequenced [18], and 75% of disease-related genes in humans have functional orthologs in the fly [19]. The *Drosophila* genetic, developmental, and biochemical blueprints for maintenance of gut epithelial stability and homeostasis are largely conserved between flies and vertebrates (Figure 1 and Figure 2). Globally available reagents allow sophisticated manipulations of gene activity in individual intestinal cell types. Therefore, for the past twenty years, research carried out in *Drosophila* has contributed to our understanding of the conserved mechanisms that govern the physiology of the gut in humans. 

In this review, we will discuss similarities between the *Drosophila* and human intestinal epithelium and how conservation of the gut biology has allowed researchers to efficiently apply the *Drosophila* midgut as a model system to study gut physiology, regeneration, immune defenses, and homeostatic host-microbiota interactions. Moreover, we will highlight how the *Drosophila* model system has the potential to aid our research on the mechanisms that underlie devastating and complex diseases of the human intestine. Diseases of the gut involve complex genetic and environmental factors which make studying the etiological causes of these diseases difficult to determine. Thus, studies within a genetically amenable, less complex, yet functionally analogous model system such as the *Drosophila* intestinal epithelium may favor the identification of genetic markers for detecting the onset of intestinal diseases and identifying new disease markers and therapeutic targets for the prevention and treatment of pathologies, such as IBD.

## 2. Gut Physiology and Homeostasis in *Drosophila melanogaster*

The anatomical and cellular architecture of the *Drosophila* intestinal epithelium has extensively been described in several reviews over the past ten years (for a detailed review, see [21]). Here, we will summarize the results of pivotal works that have contributed to define the physiology of the *Drosophila* intestine. We will focus on critical parallels and differences between *Drosophila* and mammalian intestines to highlight the pertinence of the *Drosophila* intestine as a model to dissect mechanisms that govern intestinal epithelial health and homeostasis in humans. 

The *Drosophila* gut consists of a simple epithelium surrounded by visceral muscles, nerves, and the trachea. The epithelium of the *Drosophila* adult intestine is subdivided along the anteroposterior axis into the foregut, midgut, and hindgut (Figure 1B). Both extremities have an ectodermal origin while the middle region, indicated as midgut, has an endodermal origin. The adult midgut is further subdivided into six major anatomical regions (R0 to R5 indicated in Figure 1B) with distinct metabolic and digestive functions [22]. Detailed morphometric, histochemical, and transcriptomic approaches established that these regions can be further subdivided into fourteen sub-regions (Figure 1B) [22]. The *Drosophila* midgut epithelium is composed of four different cell types (Figure 2 right panel). Progenitor cells are intestinal stem cells (ISCs) and undifferentiated ISC daughters referred to as enteroblasts (EBs); EBs can differentiate into enterocytes (ECs) or enteroendocrine cells (EEs) in response to diverse, and still not completely defined differentiation signals (Figure 2 right panel) [23,24]. ISCs and EBs express the SNAIL family transcription factor, *escargot,* which allows researchers to identify and target these cells by using *escargot* as a genetic signature. 

ECs differ notably in morphology and function within different regions of the gut. However, most of them express the gene *Myosin31DF,* while EEs express the marker gene, *Prospero*. The ECs secrete digestive enzymes and absorb and transport nutrients, while the EEs secrete hormones regulating gut mobility and function in response to biotic and abiotic stimuli coming from the external milieu. EEs provide the first antimicrobial defense, which make them crucial elements for the *Drosophila*’s innate immune system [25,26]. Moreover, the EEs release hormonal signal and peptides that potentially control inter-organ communication such as the gut–brain axis to convey the organism nutritional status and to modulate behavior and metabolism in response to nutrient availability similarly to their mammalian orthologous [27,28].

The *Drosophila* gut is structurally and functionally analogous to the human intestine (Figure 2 left panel). The posterior midgut (R4–R5 in Figure 1B) is the most metabolically active and immune responsive region of the *Drosophila* gut and is analogous to the human small intestine, while the hindgut corresponds to the human colon [23]. The posterior midgut has a simple columnar epithelium that contains progenitor cells and is not organized in an anatomical crypt-like niche structure, as in their mammalian counterparts (Figure 2 right panel) [29]. In the *Drosophila* midgut, ECs and EEs perform antimicrobial functions that are fulfilled by specialized IECs such as goblet cells and Paneth cells in mammalian intestines [25,26]. 

Different intestinal segments carry out unique functions, and many physicochemical parameters vary accordingly, such as pH and, oxygen concentration. For example, both nutrient breakdown and enzymatic activity are strongly affected by the physicochemical properties of the lumen [30]. Specifically, mammalian digestion takes place under acidic conditions that facilitate protein breakdown and the absorption of minerals and metals [21]. In contrast, digestion in flies occurs under neutral or basic pH conditions [21]. The *Drosophila* gut is mainly neutral or mildly alkaline, but some portions become strongly acidic such as in the copper cell region (R3) and the hindgut [31,32]. These physiochemical proprieties also shape the microbial distribution, composition, and density along the gut [33,34]. Thus, in humans, along the length of the gut, microbial distribution and diversity increase (stomach 10^1^, duodenum 10^3^, jejunum 10^4^, ileum 10^7^, colon 10^12^ cells per gram) as the pH of the region decreases [35]. The upper intestine mainly supports aero- and acido-tolerant bacteria such as *Helicobacteraceae* and *Lactobacillaceae*, whereas the colon strictly supports anaerobic bacteria such as *Lachnospiraceae*, *Bacteroidaceae*, and *Prevotellaceae* [34]. The *Drosophila* gut does not have an anoxic portion, unlike the human gut; therefore, the bacterial communities found in the *Drosophila* gut are aero-tolerant species and represent a limit of this model for studies of host-microbiota interactions in humans. The most commonly found bacterial species in *Drosophila* are *Lactobacillaceae* and *Acetobacteraceae* [36,37,38], which are also sensitive to physicochemical changes of the gut. For example, preventing acidification of the intestinal lumen in the copper cell region is associated with increased bacterial abundance [31,39,40], and this process also governs changes in species composition and their regional localization, resulting in increased colonization of the posterior midgut [39,40].

The intestinal epithelium is in constant contact with the external environment, but the ISCs are partially isolated and protected from it in both *Drosophila* and humans. The crypt-like niche structures of the human small intestine and one or more carbohydrate-rich barriers, such as the secreted mucus layer [41], provide a protective environment that isolates ISCs from various external stimuli in humans. Mucus has antimicrobial roles and forms a highly charged gel that acts as a physical barrier [42]. Additionally, mucus is composed of mucin glycoproteins that have anti-bacteria property [42]. 

In *Drosophila,* the intestinal epithelium is covered by a semi-permeable structure composed of chitin and proteins, the peritrophic membrane (PM), as well as an additional thin mucus layer. The *Drosophila*’s mucus layer has been identified only recently thanks to the development of a mucosubstance staining method (periodic acid Schiff) [10]. Over 30 *Drosophila* genes have been annotated as mucin-like proteins [43], but the functional relevance of these genes or, more generally, of mucus in the gut has not been investigated yet. However, the protective role of the PM against intestinal bacterial infection has been demonstrated by the study of a loss-of-function mutant that presents a reduction of PM width and an increase in its permeability [44]. These layers together serve a similar function as the human mucus layer, whereby they separate the cellular epithelium from both bacteria and toxic compounds present in the gut lumen [44,45] (Figure 2 right panel). Unlike humans, *Drosophila* does not have a lamina propria (LP), a large layer of connective tissue that contains resident innate immune cell populations such as dendritic cells, macrophages, eosinophils and mast cells, and adaptive immune cells [46]. These immune cells play important roles in regulating intestinal immunity and commensal bacterial populations by producing cytokines and immunoglobulins of the IgA-type [46]. Since *Drosophila* lacks adaptive immunity and resident immune cell populations, these aspects of the host’s immunity defenses cannot be studied in the *Drosophila* model system. Both PM and the LP selectively allow molecules to pass through and come into contact with ECs without potentially damaging them [47]. As such, active selection in the *Drosophila* PM and the human LP and mucus determines which microbial components or metabolites can come into contact with ECs.

The structural similarity between the *Drosophila* and the mammalian gut are also maintained at physiological levels as reported in the following sections.

### 2.1. Intestinal Stem Cells and Organ Plasticity

The intestinal epithelium needs to maintain its integrity in response to external and internal stress stimuli. Intestinal epithelial stress may be caused by host metabolic and nutritional changes as well as physical, chemical, and biological damages, which often lead to EC loss. To maintain its integrity, the intestinal epithelial cells, in both *Drosophila* and human gut, are continuously replaced by new cells derived from ISCs. Paracrine factors released by adjacent cells such as differentiated ECs and visceral muscles control ISC self-renewal, proliferation, and differentiation (Figure 2 right panel). ISCs proliferate upon depletion of ECs and, in *Drosophila* midguts, depleted of their EC populations, ECs can be replaced entirely within 60 h. Likewise, the human intestinal epithelium displays a similar rate and capability for regeneration [48]. Alterations in this cellular communication affect the regenerative capability of the intestinal epithelium and may lead to loss of intestinal integrity or to uncontrolled proliferation that may cause tumor development [49].

In the intestinal epithelium of young adult *Drosophila melanogaster* and humans, under normal physiological conditions, ISCs proliferate and differentiate every nine days to maintain adequate intestinal barrier integrity and function. During aging and/or stress conditions, such as bacterial infections [50], the proliferation and differentiation of ISCs may be affected, resulting in epithelial dysplasia [39]. The activity of ISCs along the digestive tract is controlled by multiple signaling pathways which are conserved from *Drosophila* to human, such as the Wnt/β-catenin signal transduction pathway (Wingless (Wg) in *Drosophila*) and Notch signaling pathway [48]. Alterations in these signaling pathways are linked to enteric diseases such as colorectal cancer in humans and appear to cause tumor-like growth also in *Drosophila* intestinal epithelium. Studying these conserved signaling pathways using the *Drosophila* gut model system has provided insight into the precise mechanisms by which these signaling pathways are activated within different ECs to maintain epithelial health.

The conserved Wg/Wnt signaling directs ISC self-renewal proliferation and promotes the regeneration of the gut upon injury in humans and *Drosophila*. Notably, in humans, Wg/Wnt dysfunction is attributed to the development of over 80% of human colorectal carcinomas [51]. Studying the R1–R5 regions of the *Drosophila* midgut (Figure 1B), Tian et al. redefined the source and role of Wg during adult intestinal development and homeostasis [52]. Tian et al. described two levels of Wg signaling activation: a high-level of Wg signaling at the regional boundaries of the intestinal epithelium and surrounding visceral muscle and low levels through the regions themselves. These two levels of regulation create a gradient that is essential to control stem cell proliferation and cell differentiation during the development of the organism. Thanks to the genetic tools available in *Drosophila*, this study also demonstrated that ECs, but not ISCs, are the primary sites of Wg pathway activation during homeostasis. In fact, non-autonomous Wg signaling in ECs regulates JAK-STAT signaling in neighboring ISCs and thereby prevents ISC over-proliferation. This information has yet to be validated in the human intestine.

Notch signaling is essential in various developmental events in both *Drosophila* and mammals [53]. In *Drosophila*, Notch signaling plays a key role in controlling the identity of ISC progeny in the gut: high Notch activation promotes the differentiation of EBs into ECs, whereas low Notch signaling promotes the differentiation of EBs into EEs. The inhibition of Notch signaling induces ISC loss and secretory cells and EE hyperplasia in the gut of both mice and *Drosophila*, respectively, while ectopic Notch signaling promotes EC differentiation [24]. The ligand of Notch receptor, Delta, is expressed explicitly by ISCs, while the Notch receptor is expressed in both ISCs and EBs [24]. A recent study showed that bidirectional Notch signaling is required to maintain multipotency in *Drosophila* ISCs [54]. During ISC asymmetric division, leading to EE cell formation, there is specific asymmetric segregation of the EE specific protein Prospero for the basal daughter cell (future EE cell) and Par complex for the apical daughter cell (future ISC) that is not observed in EC formation. Furthermore, EE cells will express Delta to activate Notch signaling in the future ISC daughter. This newly identified bidirectional mechanism for Notch signaling may also regulate multipotency in human ISCs and other stem cells regulated by this pathway, such as common lymphoid progenitors giving rise to B cells and T cells. Notch-dependent tumorigenesis has also been described in the gut of flies [55]. Given the broad conservation of immune activity and Notch signaling in the gut of flies and mammals, and the role of the Notch pathway in mouse models of colorectal cancer [56,57], this model raises the possibility to describe the relationships between intestinal immune activity and the formation of Notch-dependent tumors in a chronic inflammatory context.

The *Drosophila* system possesses a promising tool to rapidly dissect signaling pathways that are conserved in humans. The characterization of these conserved signals such as the Notch and the Wg/Wnt signaling pathways, is fundamental to better define the role of pathways associated with pathologies of the intestine, including intestinal cancers.

### 2.2. Microbiota Composition and Manipulation in the Laboratory

All eukaryotic organisms interact with multiple microorganisms that strongly influence their physiology. Since the 1960s, *Drosophila melanogaster* has been used as a model organism to characterize host-microbial interactions in the gut [58]. It is only in the past decade that our understanding of the nature and impact of the *Drosophila* gut microbiota has dramatically increased; revealing that the *Drosophila* is an excellent model organism to utilize for the study of host-microbiota interactions [20]. 

16S RNA-sequencing experiments conducted on *Drosophila* caught in their native habitats show that flies are associated with complex bacterial populations [37,59,60], yet harbor a lower microbial diversity (3–30 species) compared to that found in the mammalian gut [61,62,63] (Figure 2 middle panel). For instance, in humans, it is estimated that up to 500 different commensal bacterial species are present in the colon [64]. Some bacterial taxa are commonly associated with fly and humans such as *Lactobacillus* species. Interestingly, these bacterial species display similar health-promoting proprieties in *Drosophila* and mammals [65]. In both *Drosophila* and humans, gut microbiota abundance and diversity are more dependent on the host’s diet [66], gut morphology, and health status than on the gut’s chronological age [67]. Microbiota diversity is also influenced by the host’s genotype [68,69,70].

The *Drosophila* model presents several advantages to dissect host-microbiota interactions. Similar to the genomics era being initiated with the assessment of small and simple genomes, microbiota research could benefit from the simplified community structure present in the *Drosophila melanogaster* model to deconstruct complex polymicrobial interactions *in vivo* [20]. Different methods can be used to generate sterile (also referred to as axenic) *Drosophila* strains and rear them under germ-free conditions [71], providing the ability to carefully control environmental and experimental conditions before inoculating the flies with bacterial species of interest [72] (see Box 1). Additionally, the vast arsenal of genetic tools available, along with over 200 fully sequenced isogenic fly lines, offers the possibility to utilize genetic screens to identify and analyze host factors affected by an organism’s microbiota and host factors that are important for the colonization of the microbes [72]. 

For several years, multiple studies from independent groups have accumulated evidence that when reared in laboratory conditions, *Drosophila* lacks a stable bacterial gut population [37,59,68,73,74,75]. The absence of a “core microbiota” strain strongly suggests that there is no stable symbiont in the *Drosophila* digestive tract [37,59,68,73,74,75]. Therefore, host-microbiota research in the *Drosophila* model was limited to working with transient bacterial colonization. This limitation has been recently resolved due to the work of Pais et al., which presents rigorous and comprehensive evidence that stability within the microbial community and gut residency do take place in the digestive tract of *Drosophila melanogaster* [36]. Moreover, this work demonstrates that bacteria can colonize the *Drosophila* gut in a host-specific fashion, since bacteria that can colonize *Drosophila melanogaster* cannot colonize the intestine of the close-related species *Drosophila simulans* [36]. Additionally, colonization of the gut is dependent on the host’s genome, since specific *Drosophila* mutant strains are colonized by bacteria more efficiently than wild-derived *Drosophila* strains [36]. This discovery opens a new field of research that uses *Drosophila* as a model organism to understand the mechanisms behind gut microbiota persistence and proliferation [76].

To study host-microbiota interactions in the *Drosophila* model, researchers have several options to set-up their experiments, depending on the biological questions they wish to answer (Box 1). Different methods have been described to generate sterile axenic *Drosophila* strains and rear them under germ-free conditions [71] to determine the commensal factors that influence host traits. This experimental set-up has also been used to define what bacteria can be successfully re-introduced into the host [77]. Moreover, axenic flies can be inoculated with bacterial species of interest to generate transiently colonized [77] or stably colonized [36] flies with a defined microbial community (i.e., gnotobiotic flies). 

Comparative studies on the transcriptome of germ-free and conventionally reared *Drosophila* showed that 152 genes were differentially regulated exclusively in the gut [78,79], providing evidence that the microbiota influences host physiology under different circumstances [80]. The *Drosophila* gut microbiota, as with human microbiota, can be shaped by the diet, [81], this parameter must also be monitored to study diet–microbiota–host interactions which play a key role in neuronal development via the gut–brain axis. The ongoing optimization for a maximum standardization of the *Drosophila* culture media will make the fruit fly model an extremely valuable genetic model where both genetics and environment (e.g., nutrients and microbes) can be tightly regulated to dissect the key factors of the diet–microbiota–gut axis that affect organism health. This would be very important to translate the effect of the diet–microbiota interaction to study specific human pathologies. For instance, neurological disorders from the autism spectrum are concomitant with changes in the microbiome and enterocolitis [82]; however, the mechanisms that regulate the link between diet and/or microbiota changes with the severity of the autism symptoms are still mostly unknown. Additionally, a recent study confirmed the long-standing hypothesis that the microbiota is a crucial source of nutrients, specifically vitamins (notably, vitamin B1, also known as thiamine) for the host [83]. In this study, authors demonstrated that the host microbiota, specifically *Acetobacter pomorum*, provides thiamine to its host. Moreover, endogenous *A. pomorum* is such a main source of thiamine that flies were capable of developing on a thiamine-free diet [83].

In conclusion, the use of the *Drosophila* intestine as a model to study the bacterial microbiota has helped to define the plasticity, flexibility, and mutual benefits of host–bacteria interactions within the gut and its use will continue to help in the future to understand how the microbiota can affect different aspects of an organism’s physiology such as growth, maturation, immune and stress responses, stem cell activity, metabolic reprogramming, and disease susceptibility.

### 2.3. Pathogenic Bacterial Interactions with the Host Gut

As in all mammals, *Drosophila melanogaster* ingests potential pathogenic microbes during each meal. To defend against these pathogens, *Drosophila* has several constitutive as well as inducible levels of defense. Resistance and resilience to microbes are two critical aspects of a host’s defense [84]. In *Drosophila*, resistance to pathogens is mediated by two inducible immune response pathways: the reactive oxygen species (ROS)-generating Dual oxidase (DUOX) pathway and the antimicrobial peptide-producing immune deficiency (Imd) pathway (see the next section) [10]. Resilience involves repairing damages directly inflicted by the invading pathogen or indirect damages caused by a host’s robust immune responses. For instance, ECs damaged by pathogen virulence factors or microbicide reactive oxygen species are replaced by an enhanced compensatory proliferation of ISCs [10]. 

Due to the well-conserved mechanisms behind resistance and resilience in *Drosophila* and humans, the fruit fly has been successfully used to study infections from over 40 human pathogens, to investigate pathogen virulence factors, and how the host’s innate immune system detects and combats these microbes (for details on the contributions of the *Drosophila* model, see [85]). It appears that many human infectious agents can be effectively studied in *Drosophila*, in cases where the pathologies exhibited in flies reflected conserved aspects of human disease or physiology [85]. For example, the *Drosophila* has been used as an in vivo model to define the contribution of the type-six secretion system (T6SS) to *Vibrio cholerae* pathogenesis and assess consequences on host viability [86]. This in vivo setting was essential since *V. cholerae* pathogenicity is dependent on T6SS and host–commensal interactions [86]. 

Similarly, work in a *Drosophila* model revealed that *Pseudomonas aeruginosa*, a major agent of lethal infections in cystic and burn wound patients, modulates the local host defense responses in a tissue-dependent manner and may contribute to epithelial inflammation and cancer [87]. Moreover, *Drosophila* studies show that there is an inverse correlation between biofilm formation [88] and acute virulence and the ability of other microbial species to enhance *P. aeruginosa* virulence [89].

Moreover, studies using the *Drosophila* model system led to the discovery of a rapid and evolutionarily conserved defense response to pore-forming toxins in the gut [90]. *Drosophila* intestines exposed to hemolysin, a pore-forming toxin secreted by *Serratia marcescens*, exhibit epithelial thinning followed by a recovery of their initial thickness once the infection is cleared. A few hours post-infection, ECs expel most of their cytoplasm and organelles, although the cells do not lyse [90]. Consistent with the *Drosophila* model, epithelial thinning appears to be a fast and efficient recovery mechanism to combat intestinal infections with pore-forming toxins in mice [90], suggesting that these mechanisms could be conserved in humans. An additional mechanism of gut defense independent of the immune system occurs rapidly after the ingestion of opportunistic bacteria that secrete pore-forming toxins [91]. This previously unexplored defense mechanism involves the quick expulsion of ingested bacteria by provoking strong visceral muscle contractions in response to the DH31 neuropeptide that is released by EEs [91]. Interestingly, the human homolog of DH31 exhibited a similar function [91], suggesting that this immune response against pore-forming toxins exists in humans. 

## 3. The Detection of Bacterial Bioproducts by Enterocytes

Given that the intestinal epithelium acts as a barrier against the external environment, it consistently enters into contact with bacteria that can be beneficial or harmful to the host. Therefore, epithelial cells have developed mechanisms to discriminate between beneficial and pathogenic microbes, thus tolerating the proliferation of symbiotic microorganisms while mounting immune responses towards invading pathogens.

### 3.1. Peptidoglycan Detection in Drosophila melanogaster’s Gut

Peptidoglycan (PGN) was the first bacterial component characterized in *Drosophila* as an immune elicitor [92]. PGN is a major constituent of the bacterial cell wall where it serves a structural role in maintaining its integrity. PGN forms a crystal lattice structure composed of linear chains of two alternating amino sugars (N-acetylglucosamine and N-acetylmuramic acid) cross-linked by short peptide chains, denoted stem peptides. The glycan chain is relatively well conserved among all bacteria. However, the third residue of the stem peptide is an important distinctive feature dividing most Gram-positive and Gram-negative bacteria [93]. Typically, Gram-negative bacteria and bacilli expose *meso*-diaminopimelic acid-type PGN (Dap-PGN), while Gram-positive bacteria have a L-lysine-type PGN (Lys-PGN). 

PGN synthesis is highly coordinated with the process of cell wall remodeling to ensure growth and cell division. After each round of cell division, PGN fragments are released into the surrounding medium under the action of hydrolase enzymes [94]. Some PGN, depending on the species, will be recycled to limit the energetic costs of cell wall remodeling and evade immune detection by the host [95]. Therefore, these PGN fragments do not have the same biological effect on the host and are not recognized by the same receptors. 

In both vertebrates and invertebrates, PGN detection is mediated by the PeptidoGlycan Recognition Receptors family (PGRP), which is characterized by a functional domain that is approximately 165 amino acids in length at the C-terminus position and is denoted the PGRP domain [96]. 

In *Drosophila*, each type of PGN (Lys or Dap) will activate a specific Nuclear factor κB (NF-κB)-dependent signaling cascade [92] although, in humans, the occurrence of this distinction is not as well compartmentalized. The *Drosophila* Toll pathway is activated upon recognition of Lys-PGN and triggers an immune response in many *Drosophila* cells, but it is not activated within ECs [97]. In contrast, upon Dap-PGN recognition by Peptidoglycan-recognition proteins (PGRPs), the Imd pathway mounts an immune response in virtually every cell, including ECs [98]. The Imd pathway is the ortholog of the mammalian Tumor-Necrosis Factor pathway (TNF-α) [99]. Once activated, the Imd pathway leads to the nuclear translocation of a nuclear factor kappa-light-chain-enhancer of activated B cells (NF-κB)-like-factor (called Relish in *Drosophila*) to induce the production of antimicrobial peptides, which kill invading pathogens.

Studies using genetic, biochemical, and cellular biology experiments revealed that two PGRP receptors can detect at least two forms of PGN [100]. Peptidoglycan-recognition protein LC (PGRP-LC), the first receptor discovered, acts upstream of the Imd signaling pathway and is essential for *Escherichia coli* phagocytosis [101,102,103]. PGRP-LC is a type 2 transmembrane protein that binds PGN with its extracellular domain and initiates the Imd signaling cascade with its intracellular domain [104]. In vivo, PGRP-LC has an essential role in Imd pathway activation in tissues such as the fat body (functional homolog of the mammalian liver), hemocytes (macrophage-like cells), trachea, the ectodermic regions of the gut (proventriculus (R0) and hindgut) and in a small part of the midgut (the equivalent of the small intestine in human) together with Peptidoglycan-recognition protein LE (PGRP-LE).

Box 1Control of environmental factors in the Drosophila model system.Several stringent parameters in *Drosophila* husbandry can be controlled and implanted to minimize variation during bacterial infection or host–bacteria interaction studies:**Age:** Typically, 4–14-day-old adults are used since the final steps of gut maturation take place during the first 4 days after eclosion [105], while after 14 days, age-induced decline changes of gut physiology can be observed [106].**Gender and mating status:** Sex-specific differences have been observed in biological processes such as immune and metabolic pathways [107,108,109] and some form of IBD prevalence like Crohn′s disease [110,111]. As with humans, the *Drosophila* model presents a difference in gut plasticity between females and males [21], control of the sex as well as of the reproductive state of the flies needs to be considered in any study.**Population size:** As ethical considerations do not restrict Drosophilists, a sufficiently large numbers of individuals can and should be scored. Standard density cohorts have been determined according to physiological assay [71,112].**Diet:** As observed in the human microbiota, the *Drosophila* gut microbiota can be shaped by the diet [81]. Moreover, nutritional status influences intestinal immunity [113,114]. A holidic diet for *Drosophila*, in which the exact composition and concentration of every ingredient is known and can be manipulated, is available [115] and is available for immune and metabolic studies [116].**Microbes and immune elicitors:** The *Drosophila* model can be used as the model of 40 infection conditions observed in humans [85]. Both human pathogens and entomopathogenic microbes have been used in the study of immunity [112,117,118]. Live pathogens, purified elicitors, or heat-killed microbes can be used to separate the different aspects of the host response [118].**Time and dose of inoculation:** Standard culture conditions, typical dose, and usual route of infection are available for infection study [118]. For host-microbiota interaction studies, flies can be inoculated with bacterial species of interest to generate transiently colonized [77] or stably colonized [36] flies with a defined microbial community (i.e., gnotobiotic flies).**Temperature:** The temperature influences biochemical reactions and biological processes of all living organisms [71,112]. The three typical temperatures used in *Drosophila* studies are 18, 25 and 29 °C and flies can be maintained at a constant temperature or exposed at transient temperature treatments. The optimal condition for *Drosophila* physiology is 25 °C; lower and higher temperatures will be considered as stressful conditions for the animal physiology [119,120]. Thus, the temperature will influence the host as well as bacteria physiology. Therefore, it has to be controlled and chosen according to the biological question asked.

The PGRP-LE receptor is considered to be the main PGN sensor within the midgut [121,122]. The full-length isoform of PGRP-LE does not harbor any identifiable transmembrane domains or secreted peptide signals, so it is predicted to be a cytoplasmic receptor [123]. Unlike PGRP-LC, crystal structure data and in vivo data demonstrated that PGRP-LE can only detect monomeric PGN denoted as the tracheal cytotoxin (TCT) [104]. How TCT released by extracellular bacteria enters into the cells to interact with the receptor, PGRP-LE, has yet to be identified [124,125]. PGRP-LE is a functional homolog of the mammalian intracellular **nucleotide-binding oligomerization domain** (NOD) receptors (Figure 3) [100]. For instance, similar to NOD2, PGRP-LE is essential for the detection and autophagy-mediated elimination of the endobacteria, *Listeria monocytogenes* [126]. Mutation of the *NOD2* receptor gene in humans is linked to the development of severe forms of Crohn’s disease due to inappropriate activation of NF-κB that in turn leads to chronic inflammation in the intestinal epithelium [127]. Since the mechanism of action for NOD2 is still controversial, genetic studies aimed to determine the signaling network of PGRP-LE and its role in the modulation of inflammation in the *Drosophila* midgut could help to define the mechanisms behind chronic inflammatory enteric disorders within humans.

An in vivo genetic study of the mutant flies carrying a loss of function mutations of PGRP-LC or PGRP-LE genes revealed that PGN detection is regionalized along the midgut anterior–posterior axis. PGRP-LC works in the anterior domains (R0–R1) while PGRP-LE works in the posterior domains (R3–R5) and both receptors function in R2 [121]. Even if this model does not explain all aspects of the complex relationships between bacteria and the intestinal epithelium, it does demonstrate that *Drosophila* has developed a versatile system to modulate its immune responses. Differences among PGRP receptor distribution and alternate regulatory strategies have evolved in *Drosophila* to modulate the intensity of immune responses and to promote host survival [128,129]. To add further layers of regulation to the immune response, a regulatory isoform of PGRP-LC, denoted rPGRP-LC, is required to resolve Imd pathway activation and minimize potential tissue damage induced by the immune response. rPGRP-LC will induce endosomal degradation of the PGRP-LC receptor in response to both polymeric PGN and dead bacteria; thus, the authors proposed that polymeric PGN is a signature of dead bacteria [130]. 

Experimental evidence demonstrates that PGN acts as an elicitor across eukaryotes [131]. PGN is the mediator of both pathogenic and symbiotic bacterial interactions with both local and systemic influence on host physiology. Within humans, PGN is responsible for the pathogenicity of several bacteria [132]. For example, TCT released by *Bordetella pertussis* is the specific cytotoxic factor that induces damage to ciliated epithelial cells and inhibits neutrophil function [133]. Locally, PGN contributes to the development and maturation of isolated lymphoid follicles in the intestinal epithelium [134]. The formation of these structures is induced by intestinal bacterial-PGN upon specific activation of the NOD1 receptor and its induction of the NF-κB pathway. As expected, the absence of intestinal lymphoid follicle development in germ-free mice can be rescued by oral supplementation of TCT. The systemic impact of PGN was also reported in mouse, whereby PGN from gut bacteria microbiota could systemically prime the innate immune system via activation of the NOD1 receptor in bone marrow-derived neutrophils enhancing the elimination of two major pathogens, *Streptococcus pneumoniae*, and *Staphylococcus aureus* [135]. The PGN/NOD1/NF-κB signaling activation cascade is also linked to metabolic and inflammatory disorders such as insulin resistance [136,137].

Considering the broad range of local and systemic physiological effects that PGN has in *Drosophila* and humans, *Drosophila* is an excellent model organism to characterize novel PGN signaling functions related to innate immune surveillance, activation, and tolerance. Despite the fact that *Drosophila* shares several similarities with the mammalian detection system of PGN, future investigation is required to dissect how *Drosophila* activates and modulates intestinal inflammation in response to PGN. However, to identify unexplored immunoregulatory strategies in mammals, we also need to consider the limitation of the system. The PGN-mediated induction of immune signaling in *Drosophila* intestines is limited to the NF-κB-mediated responses through the PGRP receptor family, such as PGRP-LE, a NOD-like receptor in ECs (Figure 3). 

### 3.2. Uracil Detection in Drosophila melanogaster 

Non-commensal and commensal bacteria release PGN. Therefore, both beneficial and pathogenic bacteria are recognized by PGRP-LE/-LC. However, it is still unclear how the intestinal epithelium mounts an immune response against non-commensal bacteria while simultaneously tolerating commensal bacteria. Research by Won-Jae Lee’s group determined that bacterial-derived uracil is a hallmark of enteropathogenic bacteria in *Drosophila*, eliciting a robust immune response from the ECs [138,139,140]. Therefore, pathogens can be distinguished as uracil-positive and commensal bacteria as uracil-negative [138]. Experimental evidence demonstrated that colitogenic intestinal colonizer bacteria that carry a loss of function mutation into one gene that encodes for the bacterial UMP/uracil biosynthesis pathway change their phenotype, becoming commensal intestinal colonizers [138]. 

However, the biological significance behind uracil secretion remains to be elucidated. It has been proposed that uracil secretion may be part of the stringent response, which helps bacteria to adapt and survive the stressful conditions preceding stationary phase. Notably, uracil synthesized through the UMP/uracil biosynthesis pathway is critical for proper biofilm formation, which acts as an important virulence factor for numerous bacterial species [141]. Therefore, uracil appears to be a bacterial metabolic signature, the detection of which allows the host to monitor the presence of pathogenic bacteria within the microbiota. Currently, it is not well understood why pathogens, but not commensals, secrete uracil. Yet, it has been well established that host cells within the intestinal epithelium recognize uracil, which promotes the enzymatic activation of the dual oxidase (DUOX), a member of the nicotinamide adenine dinucleotide phosphate oxidase family (NADPH) [142]. DUOX produces microbicidal hydrogen peroxide (H_2_O_2_), which acts as a first line of defense to combat the uracil-secreting bacteria [138,142,143,144,145,146]. The generation of sufficient H_2_O_2_ production represents a fast innate immune response strategy in both *Drosophila* and mammals to combat pathogens; however, a concurrent elimination of the residual H_2_O_2_ is required to protect the host from tissue damage and inflammation [145]. Failure to balance the synthesis and elimination of H_2_O_2_ can lead to chronic epithelial inflammatory diseases [147]. To prevent an accumulation of H_2_O_2_, DUOX activity needs to be finely regulated to avoid chronic inflammation, which results in EC apoptosis and oxidative damage. Genetic studies in *Drosophila* allowed researchers to determine the signaling pathways that regulate the activation of DUOX. DUOX enzymatic activation is regulated by Hedgehog (Hh) signaling, which triggers phospholipase Cb (PLCβ)-Ca^2+^, which in turn controls the DUOX-dependent production of bactericide H_2_O_2_ [148]. Moreover, DUOX is also regulated at the transcriptional level, and the transcriptional induction of DUOX in response to infection is mediated by the MEKK1-p38 mitogen-activated protein kinases (MAPK) pathway [145,146] (Figure 4A).

Despite the extensive characterization of the DUOX pathway, host receptors that detect uracil and are responsible for DUOX activation remain unknown. Preliminary genetic screens revealed that multiple G-protein coupled receptors (GPCRs) might be involved in DUOX activation during gut–bacterial interactions. Further investigation of these GPCRs and their respective ligands will provide a more precise mechanism by which the host detects uracil and eliminates microbes via DUOX regulation in *Drosophila*. Notably, DUOX orthologues are present in the mouse and humans and have roles in innate immune responses, including intestinal epithelial immunity [150], yet their mechanisms of action remain uncharacterized. The DUOX pathway has also recently been involved in metabolic reprogramming upon enteric infection (see Section 4.2) [151]. 

In humans, the DUOX pathway contributes to mucosal immunity through a different mechanism than *Drosophila*. DUOX activation is mainly required as a source of H_2_O_2_ that is needed to catalyze the oxidative conversion of thiocyanate (SCN^−^) to hypothiocyanite (OSCN^−^) by the enzyme lactoperoxidase (Figure 4A). The production of SCN^−^ executes microbicide action within the respiratory mucosa [152]. Unfortunately, pleiotropic phenotypes exhibited by DUOX-mutant mice, such as dwarfism, make it difficult to define the role of DUOX in the murine model [153]. As a result of inadequate vertebrate models for DUOX activation, the DUOX-induced cascade has not been defined in mice or humans. 

In contrast, the amenable genetics of the *Drosophila* model has largely contributed to the characterization of the regulatory mechanisms governing DUOX activity [139]. Recent work suggests that DUOX2, an ortholog of *Drosophila* DUOX in mammals, regulates interactions between the intestinal microbiota and the mucosa to maintain immune homeostasis in mice [150]. DUOX2 is expressed in the intestinal epithelium and was found to be upregulated in patients with IBD and the corresponding mouse model [150]. Mucosal dysbiosis is a potential etiological factor that leads to IBD. Notably, mucosal dysbiosis induces DUOX2 overexpression, suggesting that DUOX2 may be a potential marker for disturbed mucosal homeostasis in patients with early-stage IBD [150].

Sources of ROS in the human GI also include NADPH oxidase found in the resident immune cells of LP, such as neutrophils and macrophages, that mediate microbial killing [154]. In *Drosophila,* macrophage-like cells do not exhibit a similar activity, although they do exhibit phagocytosis activity, and respond to H_2_O_2_ signal by migrating to the site of injury [155]. The role of macrophage-like cells in gut immunity remains to be further investigated as a recent study reveals that these cells are required for ISC proliferation after oral bacterial infection [117]. 

Due to the high level of conservation between the *Drosophila* and human signaling pathway, the discoveries coming from the studies carried in *Drosophila* will be germane for the understanding of the role of DUOX in human innate immunity.

### 3.3. Host–Commensal Interactions: Short-Chain Fatty Acids (SCFAs)

SCFAs are metabolites produced when commensal intestinal bacteria ferment compounds derived from dietary fibers digested in the duodenum [156,157]. The quality and quantity of SCFAs influence host physiology modulating the overall health and the progression of diseases, such as IBD and colorectal cancer [16]. Understanding the impact of SCFAs alone and/or in synergy with other bacterial bioproducts on host physiology is a current challenge. *Drosophila* melanogaster has helped to unveil how SCFAs affect host physiology. For example, Kamareddine et al. began to reveal the signaling pathways which respond to SCFAs and cause the host to regulate essential physiological and developmental functions of the organism [158]. Authors also demonstrated that in EEs, the Imd pathway receptor PGRP-LC recognizes the SCFA acetate and in response, activates Imd signaling, which induces the production of the endocrine peptide Tachykinin. Tachykinin regulates lipid metabolic pathways, insulin signaling, and ultimately controls *Drosophila* larval development. This study carried out in the *Drosophila* gut allowed investigators to dissect conserved signaling pathways such as the Insulin pathway modulated by metabolic interactions between the host and microbiota that impact the host’s essential physiological processes. This study and others have contributed to establishing *Drosophila* as a valuable model to research immunometabolism in higher metazoans. 

## 4. Immunometabolism in the *Drosophila melanogaster* Intestinal Epithelium

Immunometabolism is the study of interactions between immune and metabolic pathways such as glucose and fatty acid synthesis. Immunity and metabolism can reciprocally influence each other. Thus, when we talk about immunometabolism, we refer to both the effect of inflammation and immune responses which control systemic metabolism and the impacts of metabolic changes on the differentiation, activation, and regulation of the immune cells. 

Immunometabolism was first theorized in 1883 by Elie Metchnikoff [159]. Elie Metchnikoff, often considered the father of immunology, was the first to describe macrophages as cells that protect the host from pathogens and highlight their importance in the maintenance of tissue homeostasis by regulating inflammation. One century later, the discovery that the pro-inflammatory cytokine, Tumor Necrosis Factor-alpha (TNF-α), is secreted by macrophages within adipose tissue and causes insulin resistance established a direct link between immunity and metabolism [160,161,162]. Initially, immunometabolic signaling was studied to understand the basis of obesity-associated inflammations that lead to insulin resistance and glucose intolerance [163]. However, it is now clear that interactions between the immune system and metabolism are a key factor for the activation of immune cells and the regulation of inflammation. The deregulation of these interactions is associated with multiple human disorders.

The *Drosophila melanogaster* model has contributed to the advancement of our understanding of this field due to the high conservation of both metabolic and innate immune pathways that exists between *Drosophila* and vertebrates [10,164]. Similar to the conservation of innate immune pathways between *Drosophila* and humans described in the precedent sections, several metabolic pathways are also highly conserved, such as the insulin and insulin-like peptide (IIS) pathway, and the mammalian target of rapamycin pathway (mTor) [165]. For example, as in humans, the IIS pathway of flies regulates nutrient storage and growth in the animal. Moreover, insulin and immune signaling reciprocally interact in *Drosophila* as they do in humans. Notably, the activation of insulin signaling down-regulates the expression of immune genes, whereas the inactivation of insulin signaling induces immune genes expression and increases resistance to infection [166]. The *Drosophila* model presents some limitations that must be taken into consideration in each study. *Drosophila* does not have lymphocytes [167], immune cells that are very sensitive to shifts in metabolism, and also lack the somatic rearrangement or hypermutation of immune receptors associated with an acquired immune response. Unlike in mammals, insulin mutants are viable in flies, and interestingly, live longer than their wild-type counterparts [168,169]. 

In humans, acute and chronic inflammation can induce cachexia, which causes progressive weight loss due to an atypical increase in energy consumption associated with defects in carbohydrate, lipid, and protein metabolism [170]. Similarly, in adult *Drosophila*, cachexia-like phenotypes can be observed after oral infection with pathogens [163]. This phenotype is characterized by a reduction of lipid and carbohydrate associated anabolic processes such as triglyceride and glycogen syntheses, and hyperactivation of catabolic processes, such as glycolysis and lipolysis [171]. This conservation of immunometabolic interactions between the *Drosophila* and the human suggests that the use of the *Drosophila* model will help to further define immunometabolic pathways in humans.

Since this review focuses on the contribution of the *Drosophila* model system as an emerging model system to study intestinal physiology in health and disease, we will report the recent contributions to the field of immunometabolism coming from studies carried out in the *Drosophila* gut model. Since the gut is a central metabolic (e.g., nutrient absorption) and immune responsive tissue (e.g., barrier immune defense), it is a valid organ to study the cross-talks between metabolism and immunity. 

### 4.1. Immunometabolic Signaling during Enteric Infection

Recent work from Lee et al. demonstrated that bacterial uracil triggers metabolic reprogramming in ECs to induce DUOX activity and ROS production [151]. Authors found that tumor necrosis factor receptor-associated factor 3 (TRAF3) is involved in DUOX-mediated endosomal signaling and leads to mTor serine/threonine kinase inhibition upon bacterial infection. The TOR pathway plays a central role in controlling cellular metabolism, and the authors demonstrated that the inhibition of this pathway stimulates pro-catabolic signaling in EEs by decreasing lipogenesis and enhancing lipolysis. Moreover, the authors demonstrated that this metabolic reprogramming provides sufficient NADPH substrate for DUOX activity. Altogether, this data indicates that enteric pathogens modulate host metabolism at the site of infection in the *Drosophila* gut. This metabolic reprogramming is required for DUOX-dependent antimicrobial responses and host resistance against invading pathogens. The study by Lee et al. highlights the extensive metabolic modulation that intestinal bacteria can induce at the site of infection. 

Hence, studies using the *Drosophila* model system may contribute to unraveling a previously unexplored model that indicates that intestinal bacteria can also modulate host metabolism at the systemic level [80]. For example, acetate produced by commensal *Acetobacter* induces systemic insulin signaling activation that is necessary for host developmental and metabolic homeostasis [172]. However, during pathogenic *V. cholerae* infection, pathogens and host cells compete for the *Acetobacter*-derived acetate [173]. *V. cholerae* will adsorb acetate more efficiently than the host, and therefore cause systemic insulin signaling inactivation that triggers a cachexia-like phenotype [172]. SCFAs have a similar effect on metabolic mechanisms in humans that affect both intestinal and host physiology [174].

### 4.2. The Essential Role of Peroxisomes as Immunometabolic Organelles in Intestinal Epithelial Integrity and Homeostasis 

Emerging evidence demonstrating the complexity of host–bacteria interactions raises questions regarding how these finely tuned pathways are integrated into IECs to maintain enteric stability and homeostasis. 

Peroxisomes are ubiquitous organelles conserved across the breadth of eukaryotes and govern essential metabolic reactions such as cellular redox homeostasis and lipid oxidation [175,176]. Recently, it was reported that peroxisomes are involved in the regulation of immune defenses and survival of the host during microbial infection [177], suggesting that this organelle acts as an immunometabolic hub within cells.

The same group reported that peroxisomes are required to maintain intestinal epithelium homeostasis and renewal in *Drosophila* as well as to promote survival and development of the organism [149] (Figure 4B). Using the amenable genetics of *Drosophila*, the authors generated flies with dysfunctional peroxisomes in ECs, allowing them to study the cell-specific role of peroxisomes in ECs. ECs with dysfunctional peroxisomes present an accumulation of non-esterified free fatty acids and high amounts of hydrogen peroxide, which becomes exacerbated in *Drosophila* challenged with the opportunistic but not lethal bacterium *Erwinia carotovora carotovora* (*Ecc15*). These unbalanced metabolic conditions cause the induction of Tor kinase-dependent autophagy, which leads to dysplasia of the intestinal epithelium and increased apoptosis of ECs. Moreover, the lack of functional peroxisomes in ECs induces dysbiosis and compromises the immune response, effectively reducing an organism’s survival upon microbial pathogenic stress. The work by Di Cara et al. suggests that peroxisomes, act as hubs that coordinate responses from stress, metabolic, and immune signaling pathways to maintain enteric health and beneficial interactions between commensals and the host. 

Given that peroxisomes, peroxisomal functions, and AMPK-Tor autophagic signaling are conserved across the breath of eukaryotes, studies in the *Drosophila* gut are pertinent to understand how peroxisomes contribute to the regulation of immune metabolic pathways in the intestinal epithelium. Alterations in immune and metabolic signaling pathways such as autophagy, the NF-κB pathway, and AMPK signaling, are associated with the insurgence of inflammatory diseases of the intestine such as IBD. Using *Drosophila* as a model system, authors discovered that the peroxisome may play a key role in regulating IEC functions. However, the role of peroxisomes within human intestines has not yet been defined. Notably, peroxisomes are abundant in gut epithelial cells in mammals as in *Drosophila*. Fatal GI bleeding has been reported as a cause of mortality in a sub-cohort of patients affected by severe metabolic and developmental disorders caused by peroxisome dysfunction and known as Peroxisome Biogenesis Disorders [178]. Therefore, peroxisomes are essential for intestinal epithelial health, and peroxisome dysfunction contributes to disease. Follow-up studies in *Drosophila* as well as in mammalian model systems will shed light on this poorly studied organelle and its role in the maintenance of intestinal epithelial integrity and function. 

## 5. Conclusions

Although there remain differences between the physiology of the *Drosophila melanogaster* and human intestinal epithelium, the literature reviewed here strongly suggests that metabolic, immune and immunometabolic pathways that maintain gut homeostasis, integrity and functions are highly conserved between fly and humans. 

The genetic and biochemical blueprint of intestinal structure and functions is well conserved (Figure 2) in *Drosophila* and humans (Figure 1) [10,164]. Moreover, intestinal signaling pathways that control and modulate host–bacterial interactions are also conserved [10,179]. 

Both *Drosophila* and humans harbor a bacterial microbiota that the composition and diversity of which depends on the host’s genotype and environmental changes, such as the host’s diet [20]. The principal members of the microbiota are from the Firmicutes phyla in both *Drosophila* and humans (Figure 2 middle panel). 

Due to these similarities, the *Drosophila* model system has greatly contributed to the characterization of the pathways controlling intestinal immunity, gut regeneration, and homeostasis. Dysfunction of these conserved pathways results in pathological phenotypes in *Drosophila* that are similar to those exhibited in human diseases such as in IBD and intestinal cancers [179]. 

As we have highlighted in this review, *Drosophila* has allowed researchers to dissect novel immunometabolic mechanisms that control the immune function of the gut and in turn, govern intestinal homeostasis. For example, the *Drosophila* gut model contributed to the discovery of the role of peroxisomes in the intestinal epithelial cells, which is a crucial step forward in understanding their role in patients affected by Peroxisome Biogenesis Disorders who exhibit symptoms of gastrointestinal bleeding [178]. This work carried out in *Drosophila* revealed that peroxisomes maintain intestinal epithelial health [149], thus opening an avenue of investigation to determine their role in the (patho)physiology of the gut in humans.

Notably, *Drosophila* lacks an adaptive immune system and relies solely on general mechanisms of innate immunity for its immune defenses [180]. These innate immune mechanisms, which are conserved throughout mammals, play a crucial role in host health and disease prevention and have often been identified in *Drosophila* prior to other mammalian models (see the discovery of Toll receptor family, Nobel Prize 2011). Therefore, *Drosophila melanogaster* is undoubtfully an established and powerful model organism with strong potential to increasingly contribute to the elucidation of the cellular mechanisms that govern intestinal health and host–microbial interactions and is a valuable genetic platform to accelerate further discoveries, which can then be validated in mammalian models and humans.

## Figures and Tables

**Figure 1 microorganisms-07-00336-f001:**
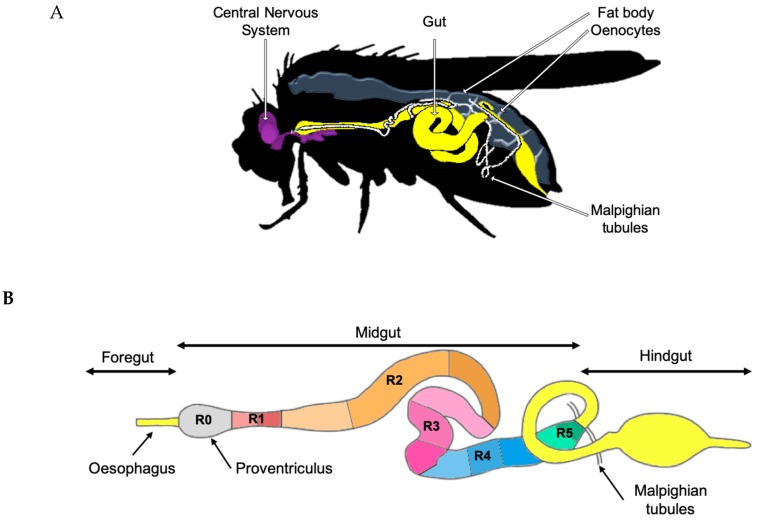
Adult *Drosophila melanogaster* anatomy. (**A**) The *Drosophila melanogaster* model system contains tissues that functionally correspond to most essential human organs: central nervous system, gastrointestinal system, adipose tissue and the liver (synergic function of fat body and oenocytes) and kidneys (Malpighian tubules). (**B**) The *Drosophila melanogaster* adult foregut, midgut, and hindgut. The midgut consists of six major anatomical regions (R0–R5) which are further subdivided into 14 color-coded sub-regions (for example, R2 is subdivided into three orange sub-regions) according to morphometric, histochemical and transcriptomic data.

**Figure 2 microorganisms-07-00336-f002:**
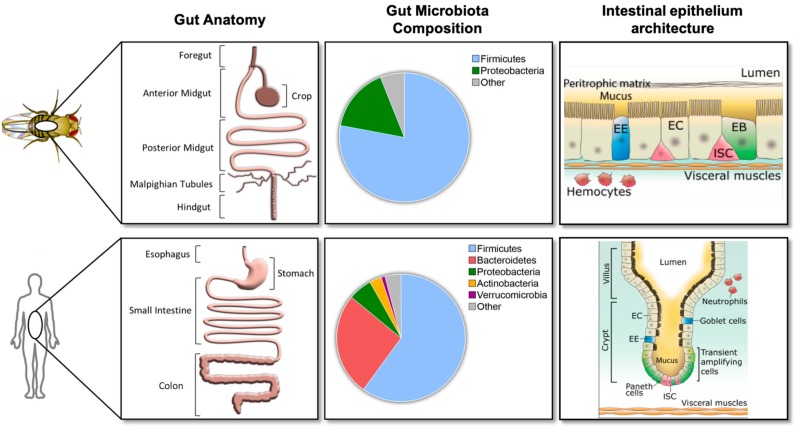
Comparisons between *Drosophila melanogaster* and human gut anatomy, gut bacterial microbiota, and intestinal epithelium. Left panel: Gut anatomy of *Drosophila melanogaster* and humans. *Drosophila* midgut and hindgut are functional analogs of the human small intestine and colon, respectively (extracted and modified from Figure 1 of [20]). Middle panel: Taxonomical distribution data for top phyla in *Drosophila melanogaster* and humans (extracted from Figure 1 of [20]). Right panel: Intestinal epithelium is surrounded by visceral muscles and composed of intestinal stem cells (ISCs) in red, undifferentiated ISC daughters in green (enteroblasts (EBs)), enterocytes (ECs) in beige and enteroendocrine cells (EEs) in blue. Under normal conditions, the gut microbiota is localized within the lumen. Barriers such as the peritrophic membrane within *Drosophila melanogaster* and the mucus within humans prevent direct contact between intestinal epithelial cells and gut bacteria.

**Figure 3 microorganisms-07-00336-f003:**
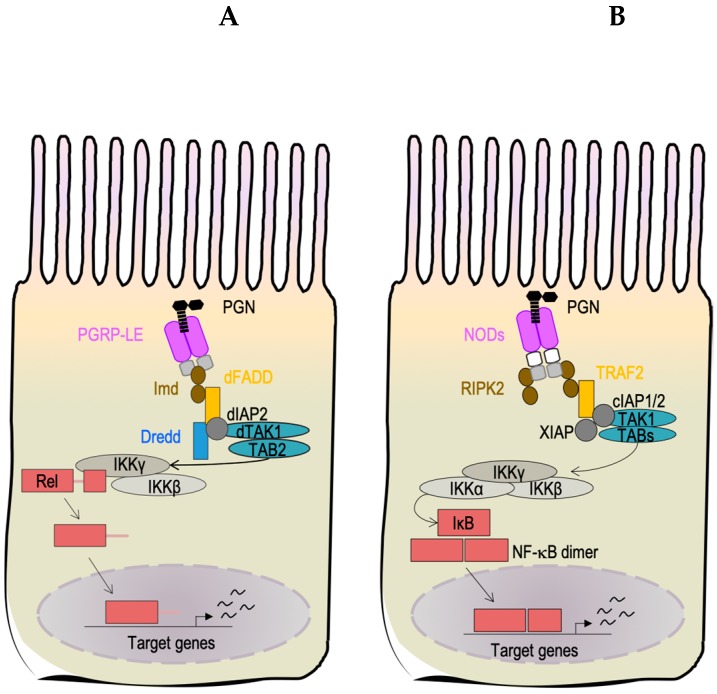
Intracellular peptidoglycan (PGN) is detected by intracellular receptors and activates the NF-κB signaling pathway in *Drosophila melanogaster* and human enterocytes. Intracellular PGN is detected by the PGRP-LE receptor in *Drosophila melanogaster* (**A**) or by NOD receptors in human (**B**). PGN-receptor interaction triggers the NF-κB signaling cascade (Imd pathway in *Drosophila*), which results in the production of antimicrobial peptides (AMPs). The orthologous proteins are represented by the same shape, color and position in the pathways.

**Figure 4 microorganisms-07-00336-f004:**
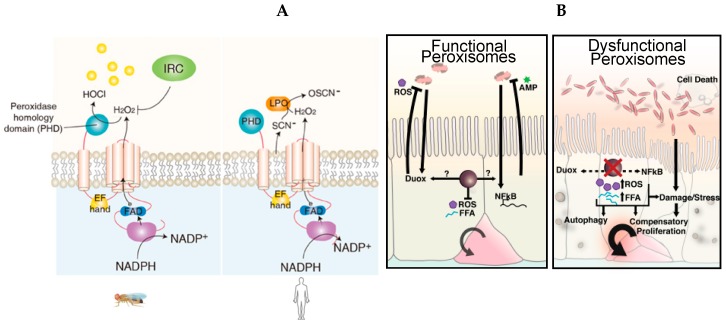
The dual oxidase (DUOX) pathway and reactive oxygen species (ROS) regulation. (**A**) Diagram of the conserved DUOX protein domains in *Drosophila melanogaster* and humans. In *Drosophila*, the peroxidase homology domain of DUOX converts H_2_O_2_ into HOCl in the presence of chloride. DUOX-dependent H_2_O_2_ molecules are eliminated by immune-regulated catalase (IRC) activity. In humans, DUOX-dependent H_2_O_2_ is used for the oxidative conversion of SCN^−^ to OSCN^−^ by the enzymatic action of lactoperoxidase in the mucosal fluids (extracted and modified from Figure 1A of [139]). (**B**) In *Drosophila* enterocytes, peroxisomes integrate and modulate the stress and immune pathways to maintain enteric homeostasis, mount host defense in the gut, and promote gut renewal. These processes are impaired in the presence of dysfunctional peroxisomes and lead to an intracellular accumulation of ROS and free fatty acids (FAAs), and cell death [149].

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
