# Peer review of "The Intestine of Drosophila melanogaster: An Emerging Versatile Model System to Study Intestinal Epithelial Homeostasis and Host-Microbial Interactions in Humans"

_microorganisms, 2019, doi:10.3390/microorganisms7090336_

Round 1

Reviewer 1 Report

The utilisation of model systems for microbiota-host interaction that have a higher throughput and flexibilty than recolonized gnotobiotic mouse is necessary especially for functional analyses.

Minor comments:

1.       Two decades of research is not short, but more important is the effort and the ressources put into a topic. Please specify these paramters.

2.       Be careful with making fruit flies a model organism for human pathology. You have to delve into the very details of pathways in order to do so.

3.       In the first two paragraphs of the introduction the references to the mentioned facts are missing.

4.       Naming human disease in a context of facts derived from fly studies is not appropriated. Please provide either specific translation from studies in flies to human pathology or avoid this context-dropping.

5.       You should define the focus of the review already in the abstract.

6.       On page 4, line 143 is awkward.

7.       What is known about the mucus layer in drosophila?

8.       Many other crucial parameters are not mentioned neither like passage time, oxygen concentration, volume.

9.       Are MAIT cells present in drosophila?

Major comments:

1.       Either follow the concept of reviewing all studies on the intestine of flies (the best choice if the number of studies is limited in numbers but broad in scope) or specify on a specific aspect if enough studies are available for this topic. You can also choose a variation of the latter strategy and use the concept of microbiota-host interaction.

2.       The description of metabolic functions in different sections the gut is much too sparse. Please refer to uptake of the different classes of molecules as well as classical mediators between host and microbes like bile acids.

3.       The lack of a core microbiome renders all colonisation experiments rather unreliable. Please provide a framework of conditions that should be taken into account in order to make future studies more reliable and comparable.

4.       How can drosophila be an optimal model to study infections from human pathogens when the overall conditions (passage time, oxygen, etc) as well as the microbiota are so different in composition and diversity?

Author Response

Thank you for your time in reviewing our manuscript.

Please find attached our point to point rebuttal letter.

Reviewer 2 Report

This review by Capo et al is very well written to summarize current knowledge of the structure and functions of the Drosophila melanogaster intestine in support of it being a valid model system to study the human intestine. The authors are apparently well versed in the anatomy, anti-bacterial mechanisms as well as associated signaling pathways of the gut of Drosophila, and present convincing evidence to demonstrate the intestine of Drosophila as an excellent model system to study immunometabolism. Overall, in my opinion, the manuscript could be improved by more elaboration of the human intestine in order to better compare and contrast the Drosophila and human guts, by providing more information on the points listed below –

Lines 163-164, in describing Drosophila lacking a lamina propria, it would be helpful to provide a brief description of the lamina propria in the human gut, such as the types of resident immune cells present, IgA antibodies, M cells, and a mucus layer. Lines 238-241, the fact that the Drosophila gut has a far less diverse gut microbiota is of a concern to researchers who study the human intestine and gut microbiome. A more in-depth discussion on what accounts for the simpler gut microbiota in Drosophila, and conditions within the gut microenvironment that are critical for the colonization of many bacteria in the human gut (pH, oxygen content, mucus layer, inter-species interactions among bacteria, etc). Given this, it would be helpful to offer a brief paragraph summarizing the limitations of the Drosophila system for the study of the colonization of many human gut bacteria. The DUOX pathway and ROS regulation system are largely conserved in the human gut. But additionally human immune cells such as neutrophils and macrophages express the NADPH oxidase that mediates generation of ROS as well, albert in this case the ROS are mainly for microbial killing. Again, a description to help the audience better differentiate the human and Drosophila systems would be helpful.

Minor points –

Line 40, should remove “a” (last letter) since “symbiotic relationships” are plural. Lines 59-60, it’s odd to have one sentence as a paragraph here.

Author Response

Thank you for taking the time to review our manuscript.

Please find attached herein our  rebuttal letter
